# SpectrumKD: Dynamic Dataset Curation for Distribution-Aware Knowledge Distillation of Large Language Models

## Abstract

Knowledge Distillation (KD) is a critical technique for compressing large language models (LLMs) into efficient student models while preserving performance, yet its efficacy remains highly sensitive to training data quality. Current dataset curation approaches mainly focus on quality and information at the instance level, neglecting the global distribution characteristics of the entire training dataset. This oversight often results in suboptimal data selection that degrades distillation outcomes. To address this limitation, we propose SpectrumKD, a principled data curation framework that dynamically refines training datasets across epochs by leveraging the global distribution of instance difficulty. SpectrumKD constructs a difficulty spectrum over the training corpus by ranking instances based on student model evaluation, partitioning them into four distinct learning phases: Early Learning, Continuous Learning, Late Learning, and No Learning. A sliding window segmentation strategy then selects epoch-specific subsets by adaptively shifting a fixed window across the spectrum from low to high difficulty, to ensure an uniform increase in subset difficulty across training epochs. As a plug-and-play module, SpectrumKD enhances diverse white-box KD methods and model architectures with minor computational cost. Extensive experiments across multiple language model benchmarks demonstrate consistent performance gains in distilled models, with improvements observed under varied KD approaches and model families. Crucially, SpectrumKD achieves these gains without modifying core distillation algorithms, highlighting the pivotal role of dataset distribution features and data compatibility in effective LLM distillation. Our work establishes a data-centric paradigm for KD, providing both insights and tools to advance the efficiency and capability of compressed language models.

## 1 Introduction

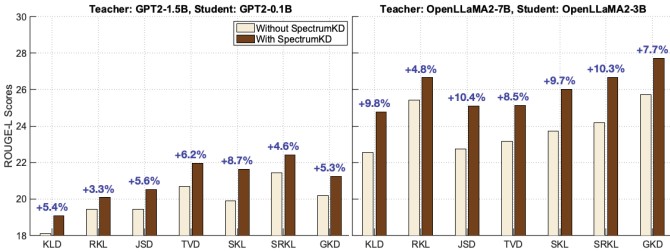

**Figure 1:** Effectiveness of the SpectrumKD framework across KD methods on five instruction-following evaluation datasets. This figure compares off-policy (KLD, RKL, JSD, TVD, SKL and SRKL) and on-policy (GKD) KD methods with and without SpectrumKD, evaluated using ROUGE-L scores. SpectrumKD improves all base white-box KD methods.

Large language models (LLMs) demand vast, high-quality data (Ouyang et al., 2022), making deployment on resource-constrained devices impractical (Aryan et al., 2023). This motivates compact models that preserve strong performance via knowledge distillation (KD) (Hinton et al., 2015). KD

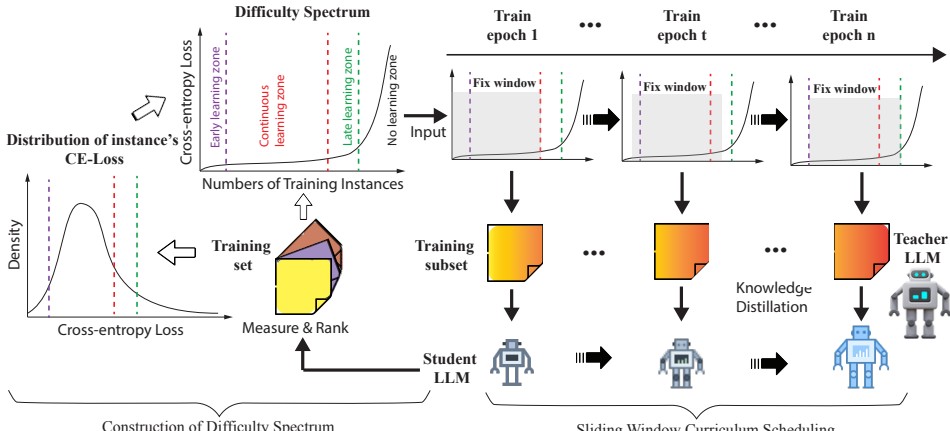

Figure 2: Overview of SpectrumKD. (1) A **difficulty spectrum** is constructed by ranking training instances by cross-entropy loss and partitioning them into four zones: Early, Continuous, Late, and No Learning. (2) A **sliding window curriculum scheduler** dynamically selects epoch-specific subsets by shifting a fixed-size window from easy to hard regions, ensuring a smooth increase in training difficulty aligned with the student's evolving capacity.

for LLMs is broadly categorized as black-box—using teacher-generated outputs for supervised fine-tuning (SFT), suitable for closed models like GPT-4o (Hurst et al., 2024)—or white-box, which exploits the teacher's full probability distribution and is feasible with open models such as DeepSeek-v3 (Liu et al., 2024) and Qwen 2.5 (Yang et al., 2024a).

White-box KD research has largely centered on distillation losses—extending beyond standard Kullback-Leibler divergence (KLD) to variants like reverse KLD RKL (Gu et al., 2023), Jensen-Shannon divergence JSD (Agarwal et al., 2024), skew KLD variants (SKL/SRKL) (Ko et al., 2024), and others (Wang et al., 2025; Shing et al., 2025). Yet these methods often lack consistent cross-task generalization (Agarwal et al., 2024; Ko et al., 2024), suggesting that loss design alone is insufficient. Crucially, while data curation is well-studied in black-box KD and SFT, it remains neglected in white-box settings. We address this gap by asking: *How can fixed datasets be dynamically curated to improve white-box distillation?* Our work focuses on two key questions: (1) **what** data to select, and (2) **when** to use it during training.

Regarding **what** training data to use, leveraging high-quality data is widely regarded as essential in KD (Ding et al., 2023; He et al.) and SFT (Li et al., 2023a; 2024). However, "high quality" remains ill-defined. Most approaches equate quality with instance difficulty—assuming harder examples are more informative. For instance, Instruction-Following Difficulty scores guide data selection in instruction tuning (Li et al., 2023a), while Distillation Difficulty Scores serve a similar role in KD (He et al.). SKD (Xu et al., 2024b) filters samples based on the distributional divergence between teacher and student, prioritizing instances where the student struggles most.

Yet recent work challenges this assumption, showing that high difficulty does not always translate to better distillation outcomes (Ko et al., 2025). The capacity gap between teacher and student models (Cho & Hariharan, 2019) can induce a training-inference mismatch (Lin et al., 2020; Agarwal et al., 2024), limiting the utility of overly challenging data. To address this, hybrid strategies combine ground-truth labels (Hinton et al., 2015), teacher generated outputs (Kim & Rush, 2016), and student generated outputs (Lin et al., 2020; Agarwal et al., 2024)—a paradigm known as on-policy distillation. These methods reveal that so-called "low-quality" samples, such as early student generations, can play a constructive role in white-box KD, suggesting that data quality must be re-evaluated in a distribution-aware and student-adaptive manner.

Concerning **when** to introduce data during training, static difficulty-based selection is suboptimal: as the student's capacity evolves, the perceived difficulty of an instance changes dynamically. On-policy methods naturally adapt to this shift, as student-generated data improves over time (Lin et al., 2020). However, they incur high computational overhead and risk propagating errors if student outputs are unreliable (Ko et al., 2024). In off-policy white-box KD—where only teacher logits are

available—a dynamic data curation strategy is more practical. Curriculum learning (CL) (Bengio et al., 2009), which sequences training from easy to hard examples, offers a promising direction. Recent attempts integrate CL into KD (Liu & Zhang, 2025a; He et al.), but they rely solely on instance-level difficulty and ignore the global distributional structure of the dataset. Without awareness of the overall data spectrum, curriculum scheduling becomes arbitrary, undermining CL's potential in distillation.

To address these challenges, we propose **SpectrumKD**, a principled, plug-and-play framework for dynamic dataset curation in white-box KD. SpectrumKD constructs a fixed *difficulty spectrum* from a given corpus by ranking instances according to their cross-entropy loss under an initial student model. The spectrum is partitioned into four zones—*Early Learning*, *Continuous Learning*, *Late Learning*, and *No Learning*—based on the empirical distribution of losses. Instances in the *Early Learning* zone (low loss) are used only in early epochs, those in *Continuous Learning* (moderate loss) are retained throughout training, *Late Learning* instances (higher loss) are introduced in later epochs, and *No Learning* instances (extreme loss) are excluded as incompatible.

To dynamically schedule data, SpectrumKD slides a fixed-size window across the spectrum from left to right over epochs, selecting an epoch-specific subset that gradually increases in difficulty (see Figure 2). This strategy ensures alignment between the student's evolving capacity and the training data, filtering out mismatched samples while intensively leveraging high-quality, compatible instances. Evaluated on instruction following, mathematical reasoning, and code generation tasks across diverse model families—including GPT-2 (Radford et al., 2019), OpenLLaMA2 (Geng & Liu, 2023), and Qwen2.5 (Team, 2024)—SpectrumKD consistently boosts distillation performance with minimal overhead (see Figure 1).

In summary, our contributions are twofold:

**Methodologically**: We introduce SpectrumKD, a distribution-aware curation framework that adaptively selects student-compatible, high-quality training subsets per epoch, enhancing flexibility and efficacy in white-box KD.

**Empirically**: Comprehensive experiments across tasks, architectures, and distillation settings demonstrate that SpectrumKD delivers consistent gains with negligible computational cost, underscoring its generality and practical utility.

## 2 BACKGROUND AND RETHINKING

### 2.1 BACKGROUND

**White-Box KD for Auto-regressive Language Models.** In white-box KD, a student model learns from a more capable teacher using a dataset of prompt-response pairs $(x, y)$. The training objective combines two components: (1) the cross-entropy loss ($L_{ce}$) between the ground-truth sequence $y$ and the student's predicted distribution $q_\theta(y|x)$, and (2) the KD loss ($L_{kd}$), which measures the divergence between the teacher's softened output distribution $p(y|x)$ and the student's distribution. The total loss is formulated as:

$$L_s = \alpha \cdot L_{ce} + (1 - \alpha) \cdot L_{kd}, \tag{1}$$

where $\alpha \in [0, 1]$ controls the trade-off between ground-truth fidelity and knowledge transfer. Specifically, $L_{ce} = -\sum_{i=1}^{|y|} \log q_\theta(y_i|x, y_{<i})$, and $L_{kd} = \sum_{i=1}^{|y|} D\left(p(y_i|x, y_{<i}; \tau) \parallel q_\theta(y_i|x, y_{<i}; \tau)\right)$, with $D(\cdot \parallel \cdot)$ denoting a divergence (e.g., KL divergence) and $\tau$ a temperature parameter that smooths the distributions. Further details are provided in Appendix A.

**Limitations of Training Dataset Utilization in White-Box KD.** A critical issue in LLM distillation is the training-inference mismatch: fixed training datasets often fail to reflect the input diversity encountered during inference (Ko et al., 2024). This mismatch stems largely from the capacity gap between teacher and student models (Cho & Hariharan, 2019), which undermines the assumption that "high-quality" data—typically defined by high teacher-student divergence or instruction difficulty (Li et al., 2023a)—is universally beneficial. In practice, such instances may be too complex for the student to learn effectively, leading to poor knowledge transfer.

Consequently, data compatibility with the student's current learning capacity deserves greater attention. Since the student's capabilities evolve during training, the notion of "high-quality" data is in-

herently dynamic. While on-policy strategies that incorporate student-generated outputs (Lin et al., 2020; Agarwal et al., 2024) improve compatibility by aligning training with inference dynamics, they introduce significant overhead—student generation can consume up to 80% of total training time (Ko et al., 2024)—and risk error propagation if noisy outputs dominate. Conversely, purely off-policy approaches using only static, pre-selected data neglect this dynamic nature, limiting distillation efficacy. These trade-offs motivate the need for a lightweight, dynamic curation mechanism that adapts to the student's evolving capacity without incurring substantial computational cost.

## 2.2 RETHINKING TRAINING DATASET FOR KD

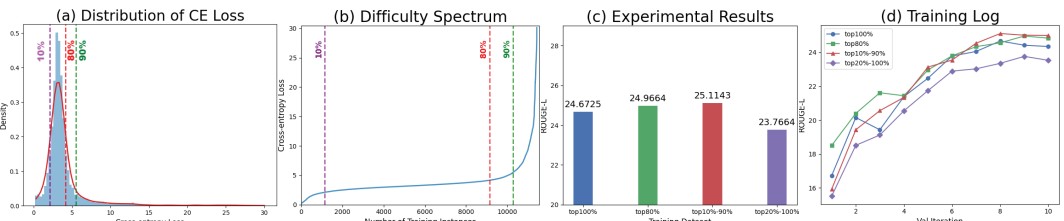

Figure 3: Empirical analysis motivating SpectrumKD. (a) Cross-entropy loss distribution over the Dolly dataset exhibits a long-tailed pattern. (b) Difficulty spectrum formed by sorting instances in ascending loss order. (c) ROUGE-L scores when training on four difficulty-based subsets. (d) Validation convergence curves for the same subsets.

**Empirical Analysis.** To investigate the interplay between data quality and student compatibility in white-box KD, we distill a GPT-2 (0.1B) student from a GPT-2 (1.5B) teacher (Radford et al., 2019) on the Dolly dataset (Conover et al., 2023); full details are in Appendix C. Using the untrained student, we compute the cross-entropy loss for each training instance, treating it as a proxy for difficulty: low loss indicates "easy" instances, high loss denotes "hard" ones.

As shown in Figure 3(a), the loss distribution follows a log-normal pattern (Akkurt et al., 2019), exhibiting a sharp peak and a long right tail—indicating that while most instances are easy, extremely hard samples occur with non-negligible frequency. Sorting instances by loss yields a difficulty spectrum (Figure 3(b)) that reflects this long-tailed structure. We then train students on subsets with varying difficulty profiles. Results in Figures 3(c)–(d) reveal three key trends: (1) training on either the easiest 80% (Top80%) or the moderate-difficulty band (Top10%–Top90%) outperforms using the full dataset; (2) the easy-only subset enables faster convergence; and (3) including extreme hard instances (e.g., Top10%) degrades performance.

These findings lead to two insights for effective data curation in white-box KD:

- *Training datasets exhibit long-tailed difficulty distributions: most instances are easy and redundant, while rare but extreme hard instances—though infrequent—can significantly impair distillation if included indiscriminately.*
- *Easy instances accelerate convergence, whereas moderate-difficulty instances enhance final performance; extreme hard instances should be excluded to avoid destabilizing training.*

These observations directly inform our dynamic curation strategy in Section 3.

**Theoretical Analysis.** Our empirical results align with theoretical analyses of teacher-student capacity gaps in KD (Ko et al., 2025). For easy samples, the student's initial output distribution $q_\theta$ is close to the teacher's $p$, yielding low KD divergence and stable gradients. In contrast, hard samples induce large distributional mismatches, producing high-variance gradients that hinder optimization—especially for extreme cases, which can actively degrade learning.

From a compatibility perspective, easy instances provide low-noise, high-fidelity supervision, facilitating stable early training. However, relying solely on them limits the student's exposure to complex patterns encoded by the teacher. Crucially, as training progresses, the student's capacity increases, rendering previously hard samples more learnable—paralleling the adaptive refinement in on-policy KD. This suggests a two-part strategy: (1) exclude extreme hard instances that exceed the student's representational capacity, and (2) adopt a curriculum-based curation approach that

aligns sample difficulty with the student's evolving competence. By progressively introducing more challenging—but still compatible—data, the student assimilates richer knowledge without abrupt distributional shifts (Wang et al., 2021).

## 3 METHODOLOGY

### 3.1 MOTIVATION

Motivated by the empirical and theoretical insights in Section 2.2, we propose a dynamic data curation framework that leverages the global difficulty spectrum of a training dataset to guide epoch-wise subset selection in knowledge distillation. Our approach begins by categorizing all instances into four types based on their cross-entropy loss under an initial student model: (1) *Early-stage Learning*, (2) *Continuous Learning*, (3) *Late-stage Learning*, and (4) *Excluded* instances.

*Early-stage Learning* instances are the easiest samples, for which the student's initial output distribution $q_\theta$ closely aligns with the teacher's $p$, yielding low distillation divergence and stable gradients—ideal for early training. *Continuous Learning* instances, constituting the majority of the dataset, exhibit moderate difficulty and high compatibility with the student throughout training. Their repeated exposure aligns with the principle of deliberate practice (Anders Ericsson, 2008), maximizing knowledge retention. *Late-stage Learning* instances are harder samples that exceed the student's initial capacity but become learnable as training progresses; introducing them later enables the student to absorb complex patterns and better exploit the teacher's knowledge. Finally, *Excluded* instances—extreme outliers with very high loss—are discarded to prevent destabilization and negative transfer.

Building on this classification, SpectrumKD dynamically constructs epoch-specific training subsets. Early epochs include only Early- and Continuous-stage instances. As training advances, Early-stage instances are gradually phased out and replaced by Late-stage ones, culminating in a subset composed solely of Continuous and Late-stage instances. By excluding the most difficult samples, the effective dataset size is reduced, allowing more training epochs within a fixed compute budget and enabling intensified focus on high-value, compatible data. We formalize this strategy in **SpectrumKD**, a two-stage framework: (1) construction of a fixed difficulty spectrum from the initial student's loss distribution, and (2) curriculum scheduling via a sliding window that progressively shifts across the spectrum to modulate data difficulty over epochs.

### 3.2 CONSTRUCTION OF DIFFICULTY SPECTRUM

**Spectrum Construction.** Given a dataset $D = \{(x_i, y_i)\}_{i=1}^N$, we evaluate each instance using an untrained student model to compute its cross-entropy loss $\mathcal{L}_i = -\log q_\theta(y_i|x_i)$. Lower loss indicates higher compatibility (i.e., "easier" instances). Sorting instances in ascending order of $\mathcal{L}_i$ yields the difficulty spectrum $D_{ds} = \{s_1, s_2, \ldots, s_N\}$, where $s_i$ denotes the $i$-th easiest sample.

**Spectrum Partitioning.** We partition $D_{ds}$ into four zones based on two configurable thresholds $\lambda_a, \lambda_b \in [0, 1]$ with $2\lambda_a + \lambda_b \leq 1$:

- *Early Learning*: $s_i$ for $i \leq \lambda_a \cdot N$   (easiest samples),
- *Continuous Learning*: $s_i$ for $\lambda_a \cdot N < i \leq (1 - \lambda_a - \lambda_b) \cdot N$   (moderate-difficulty core),
- *Late Learning*: $s_i$ for $(1 - \lambda_a - \lambda_b) \cdot N < i \leq (1 - \lambda_b) \cdot N$   (harder samples),
- *Excluded*: $s_i$ for $i > (1 - \lambda_b) \cdot N$   (extreme outliers, discarded).

The curated spectrum $D'_{ds} = \{s_i \mid i \leq (1 - \lambda_b) \cdot N\}$ forms the basis for dynamic scheduling.

### 3.3 SLIDING WINDOW CURRICULUM SCHEDULING

Given $D'_{ds}$, SpectrumKD constructs epoch-specific subsets via a sliding window that moves across the spectrum. Let the initial subset size be $M = (1 - \lambda_a - \lambda_b) \cdot N$, matching the union of Early and Continuous zones. During a warm-up phase of $k$ epochs, the model trains on the fixed subset $SD_1 = \{s_i \mid i \leq M\}$ to ensure stable early learning.

After warm-up, we employ a **sliding window curriculum scheduler**. A window of fixed size $M$ starts at the leftmost position of $D'_{ds}$ and shifts rightward over epochs. At epoch $j > k$, the window covers indices $[w_j, w_j + M - 1]$, defining subset $SD_j$. The shift step is chosen to ensure a near-uniform increase in subset difficulty. Specifically, we define the total difficulty of a subset as $td(SD_j) = \sum_{s_i \in SD_j} \mathcal{L}_i$. Let $\Delta_{\max} = td(SD_n) - td(SD_{k+1})$ be the total difficulty increase from the first post-warmup to the final epoch. We aim for an average per-epoch increase of $\Delta_{\text{avg}} = \Delta_{\max}/(n - k - 1)$. The window shift at each epoch is selected to make the actual difficulty increment as close as possible to $\Delta_{\text{avg}}$, ensuring smooth progression from easier to harder—but compatible—data.

**Adaptive Temperature Scheduling.** To further align supervision with student capacity, we linearly increase the distillation temperature: $\tau_j = \tau_0 + (\tau_n - \tau_0) \cdot \frac{j-1}{n-1}$. This encourages the student to first learn confident, teacher-aligned predictions (low $\tau$) and later absorb nuanced, softened knowledge (high $\tau$). Full algorithmic details are in Algorithm 1 (Appendix B).

## 4 EXPERIMENTS

We evaluate SpectrumKD on general instruction-following, mathematical reasoning, and code generation tasks. Key hyperparameters are set to $\lambda_a = 0.1$, $\lambda_b = 0.1$, and distillation temperature $\tau$ linearly increases from 1 to 2. The SFT weight is $\alpha = 0.3$ for off-policy KD and $\alpha = 0$ for on-policy KD. We compare against seven white-box KD baselines: GKD (on-policy) (Agarwal et al., 2024), KLD (Hinton et al., 2015), RKL (Gu et al., 2023), JSD, TVD (Wen et al., 2023), SKL, and SRKL (Ko et al., 2024). Full experimental details are in Appendix C.

### 4.1 GENERAL INSTRUCTION-FOLLOWING

**Setup.** We distill GPT-2 (0.1B student from 1.5B teacher) (Radford et al., 2019) and OpenLLaMA2 (3B from 7B) (Geng & Liu, 2023) on the `Databricks-dolly-15k` dataset (Conover et al., 2023) (11.5K train, 1K validation, 0.5K test). Models are evaluated on five benchmarks: DollyEval, SelfInst (Wang et al., 2022a), VicunaEval (Chiang et al., 2023), S-NI (Wang et al., 2022b), and UnNI (Honovich et al., 2022). We report (1) average Rouge-L scores over five seeds (generation temperature = 1), and (2) winning rates (WR) via LLM-as-a-Judge (Zheng et al., 2023) using DeepSeek-V3-0324 (Liu et al., 2024).

| | OpenLLaMA2-7B ($M_t$) $\rightarrow$ OpenLLaMA2-3B ($M_s$) | | | | | | | | | | | | | |
|---|---|---|---|---|---|---|---|---|---|---|---|---|---|---|
| | **DollyEval** | | **SelfInst** | | **VicunaEval** | | **S-NI** | | **UnNI** | | **Avg.** | | **P. Gains** | |
| KD Methods | *R-L* | *WR* | *R-L* | *WR* | *R-L* | *WR* | *R-L* | *WR* | *R-L* | *WR* | *R-L* | *WR* | *(%)* | *(%)* |
| $M_t$ | 26.26 | 52.03 | 19.15 | 51.12 | 17.31 | 49.39 | 31.20 | 51.34 | 31.84 | 51.05 | 25.15 | 50.98 | 0.00 | 0.00 |
| SFT | 24.87 | 49.20 | 18.78 | 50.29 | 16.50 | 48.99 | 28.65 | 47.87 | 28.73 | 47.61 | 23.51 | 48.79 | 0.00 | 0.00 |
| GKD | 26.30 | 50.50 | 18.91 | 50.06 | 17.53 | 50.92 | 34.21 | 52.38 | 31.67 | 50.46 | 25.72 | 50.87 | 0.00 | 0.00 |
| +SpectrumKD | **27.89** | **53.95** | **19.89** | **53.05** | 18.12 | 53.50 | **36.81** | **56.59** | **35.82** | **55.39** | **27.71** | **54.50** | 7.70 | 7.14 |
| KLD | 23.76 | 49.68 | 17.48 | 48.05 | 15.68 | 47.97 | 28.42 | 48.10 | 27.48 | 46.75 | 22.56 | 48.11 | 0.00 | 0.00 |
| +SpectrumKD | 25.02 | 49.49 | 18.32 | 51.33 | 17.02 | 51.66 | 29.31 | 50.53 | 34.12 | 53.90 | 24.76 | 51.38 | 9.72 | 6.80 |
| RKL | 26.67 | 50.78 | 18.78 | 49.71 | 18.43 | 53.67 | 31.27 | 50.42 | 32.01 | 52.18 | 25.43 | 51.35 | 0.00 | 0.00 |
| +SpectrumKD | 27.01 | 52.61 | 19.02 | 51.93 | **18.83** | 52.23 | 33.25 | 54.00 | 35.15 | 52.60 | 26.65 | 52.68 | 4.80 | 2.58 |
| JSD | 25.64 | 50.34 | 17.18 | 47.54 | 15.56 | 47.58 | 27.93 | 49.29 | 27.39 | 46.48 | 22.74 | 48.25 | 0.00 | 0.00 |
| +SpectrumKD | 26.83 | 52.98 | 18.32 | 51.24 | 17.83 | 53.14 | 29.31 | 49.36 | 33.19 | 53.45 | 25.10 | 52.03 | 10.36 | 7.85 |
| TVD | 24.62 | 48.57 | 18.32 | 48.90 | 15.85 | 48.49 | 29.01 | 48.54 | 28.08 | 47.54 | 23.18 | 48.41 | 0.00 | 0.00 |
| +SpectrumKD | 26.02 | 51.40 | 19.01 | 51.75 | 16.49 | 50.96 | 32.87 | 51.68 | 31.32 | 51.79 | 25.14 | 51.51 | 8.48 | 6.41 |
| SKL | 24.99 | 49.08 | 18.77 | 49.54 | 16.69 | 49.92 | 29.71 | 50.86 | 28.49 | 48.02 | 23.73 | 49.48 | 0.00 | 0.00 |
| +SpectrumKD | 26.42 | 52.11 | 19.53 | 52.68 | 18.39 | 53.83 | 30.41 | 49.41 | 35.34 | 54.97 | 26.02 | 52.60 | 9.64 | 6.30 |
| SRKL | 25.99 | 52.06 | 19.11 | 50.16 | 16.93 | 50.36 | 30.07 | 49.94 | 28.81 | 48.38 | 24.18 | 50.18 | 0.00 | 0.00 |
| +SpectrumKD | 27.35 | 51.35 | 19.82 | 52.89 | 18.52 | **54.04** | 34.82 | 55.15 | 32.83 | 53.08 | 26.67 | 53.30 | 10.28 | 6.22 |

Table 1: Instruction-following evaluation on OpenLLaMA2 (7B→3B). Baselines include SFT and seven white-box KD methods (GKD: on-policy with 50% student outputs; others: off-policy). Reported metrics: ROUGE-L (R-L) and winning rate (WR) across five benchmarks; Avg. = mean across datasets. Best student scores in **bold**; red indicates student outperforms teacher. Averaged over five random seeds. GPT-2 results in Table 6.

**Results.** As shown in Table 3 and Appendix Table 6, SpectrumKD consistently enhances student performance across all KD methods and model scales, confirming its robustness and scalability. Gains are especially notable with divergence-based losses, underscoring SpectrumKD's compatibility with diverse distillation objectives. Remarkably, the OpenLLaMA2 student not only outperforms its baselines but also exceeds the teacher on multiple benchmarks—likely because larger students

estimate instance difficulty more accurately, yielding a higher-fidelity difficulty spectrum. All improvements are statistically significant ($p < 0.05$, Wilcoxon signed-rank test).

## 4.2 MATHEMATICAL REASONING AND CODE GENERATION

**Setup.** For mathematical reasoning, we distill Qwen2.5-Math-1.5B from Qwen2.5-Math-7B-Inst (Yang et al., 2024b) using 20K training and 2K validation samples from MetaMathQA (Yu et al., 2023), with chain-of-thought prompting (Wei et al., 2022). Models are evaluated on GSM8K (Cobbe et al., 2021) and MATH (Hendrycks et al., 2021) using *pass@1*.

For code generation, we distill Qwen2.5-Coder-1.5B from Qwen2.5-Coder-7B-Inst (Hui et al., 2024) using 20K/2K samples from `Evol-Instruct-Code-80k-v1` (Luo et al., 2023), a dataset enhanced via Evol-Instruct (Xu et al., 2024a). Evaluation is performed on HumanEval (Chen et al., 2021) and MBPP (Austin et al., 2021) under the EvalPlus framework (Liu et al., 2023), also using *pass@1*.

| | Qwen2.5-Math-7B-Inst ($M_t$) → Qwen2.5-Math-1.5B ($M_s$) | | | Qwen2.5-Coder-7B-Inst ($M_t$) → Qwen2.5-Coder-1.5B ($M_s$) | | |
|---|---|---|---|---|---|---|
| | **GSM8K** | **MATH** | **Avg.** | **HumanEval** | **MBPP** | **Avg.** |
| KD Methods | *pass@1* | *pass@1* | *pass@1* | *pass@1* | *pass@1* | *pass@1* |
| $M_t$ | 90.3 | 80.2 | 85.3 | 61.6 | 76.9 | 69.3 |
| $M_s$ | 75.9 | 48.7 | 62.3 | 42.4 | 60.4 | 51.4 |
| GKD | 80.2 | 58.3 | 69.3 | 46.2 | 62.6 | 54.4 |
| +SpectrumKD | **81.4** | 58.6 | **70.0** | 47.1 | 62.2 | 54.7 |
| KLD | 77.9 | 55.2 | 66.6 | 45.1 | 61.3 | 53.2 |
| +SpectrumKD | 78.7 | 57.1 | 67.9 | 46.2 | 61.9 | 54.1 |
| RKL | 78.4 | 56.4 | 67.4 | 45.4 | 61.4 | 53.4 |
| +SpectrumKD | 79.9 | 57.3 | 68.6 | 45.8 | 61.7 | 53.8 |
| JSD | 78.9 | 56.8 | 67.9 | 45.1 | 61.0 | 53.1 |
| +SpectrumKD | 79.9 | 57.3 | 68.6 | 45.3 | 61.5 | 53.4 |
| TVD | 78.2 | 56.9 | 67.6 | 46.2 | 61.7 | 54.0 |
| +SpectrumKD | 79.7 | **59.7** | 69.7 | 46.8 | 62.2 | 54.5 |
| SKL | 79.4 | 57.9 | 68.7 | 46.4 | 61.8 | 54.1 |
| +SpectrumKD | 80.5 | 58.6 | 69.6 | 47.1 | 62.8 | 55.0 |
| SRKL | 79.5 | 58.4 | 69.0 | 46.8 | 62.0 | 54.4 |
| +SpectrumKD | 80.8 | 58.9 | 69.9 | **47.5** | **63.1** | **55.3** |

Table 2: Results on mathematical reasoning (GSM8K, MATH) and code generation (HumanEval, MBPP) using Qwen2.5-Math and Qwen2.5-Coder (7B→1.5B). Reported as *pass@1*; best scores in **bold**.

**Results.** As shown in Table 2, SpectrumKD consistently boosts performance across all KD baselines on both GSM8K and MATH. Similarly, in code generation, it yields uniform gains on HumanEval and MBPP. These results confirm SpectrumKD's effectiveness and generality beyond instruction following, extending to structured reasoning and code synthesis tasks.

## 5 ANALYSIS AND DISCUSSION

We conduct ablation studies on instruction-following and mathematical reasoning tasks using GPT-2 and Qwen2.5-Math to dissect the contributions of SpectrumKD's core components, assess hyperparameter sensitivity, measure computational overhead, and compare against alternative curriculum strategies.

### 5.1 ABLATION ANALYSIS

**Impact of Difficulty Metrics.** SpectrumKD relies on accurate difficulty estimation to construct its spectrum. We compare three metrics: (1) cross-entropy loss (our default), (2) ROUGE-L between student generations and ground truth (Liu & Zhang, 2025a), and (3) sentence-level entropy (Zhu et al., 2021). Using KLD as the distillation loss, we evaluate GPT-2 on five instruction-following benchmarks and Qwen2.5-Math on GSM8K and MATH.

Results (Tables 3, 7) show that cross-entropy loss and ROUGE-L perform comparably and outperform entropy in instruction following. In mathematical reasoning, however, cross-entropy loss yields the best results, while ROUGE-L performs worst—likely because ROUGE-L fails to capture critical semantic errors (e.g., a single arithmetic mistake), which are decisive in math tasks. This highlights the robustness and task-agnostic suitability of cross-entropy loss for difficulty estimation.

|  | GPT-2-1.5B ($M_t$) $\rightarrow$ GPT-2-0.1B ($M_s$) | | Qwen2.5-Math-7B-Inst ($M_t$) $\rightarrow$ Qwen2.5-Math-1.5B ($M_s$) | |
|---|---|---|---|---|
|  | **Avg.** | **P. Gains** | **Avg.** | **P. Gains** |
| KD Methods | *Rouge-L* | (%) | *pass@1* | (%) |
| KLD | 18.10 | - | 66.55 | - |
| +SpectrumKD (CE-LOSS) | 19.07 | 5.34% | 67.90 | 2.03% |
| +SpectrumKD (ROUGE-L) | 19.06 | 5.33% | 66.80 | 0.38% |
| +SpectrumKD (SE) | 18.35 | 1.37% | 67.25 | 1.05% |

Table 3: Ablation on difficulty metrics for spectrum construction: CE-LOSS (cross-entropy), ROUGE-L, and SE (sentence entropy). Avg. performance gains (P. Gains) over baseline KD (KLD) shown for GPT-2 and Qwen2.5-Math.

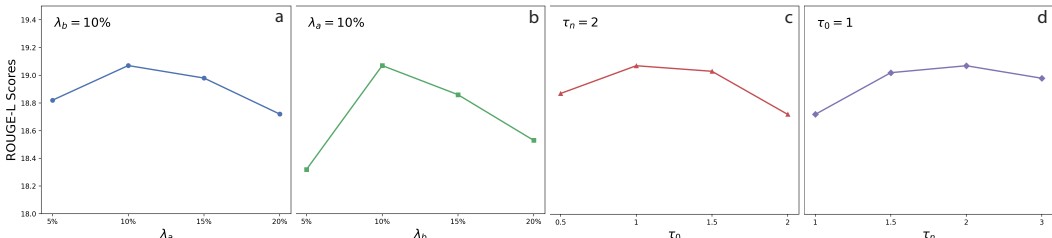

Figure 4: Hyperparameter sensitivity of SpectrumKD. Impact of (a) $\lambda_a$, (b) $\lambda_b$, (c) initial temperature $\tau_0$, and (d) final temperature $\tau_n$. Results averaged over five instruction-following datasets using GPT-2 with KLD.

**Impact of Sliding Window Strategy.** We analyze three key design choices in our curriculum scheduler: (1) forward sliding (easy $\rightarrow$ hard), (2) warm-up stage, and (3) adaptive step size. As shown in Tables 4 and 8 (Appendix D), forward sliding consistently outperforms reverse sliding (hard $\rightarrow$ easy); including a warm-up phase improves stability and final performance; and adaptive step sizing—guided by difficulty increments—surpasses fixed linear steps across both tasks and all benchmarks. These results validate the necessity of each component in SpectrumKD's scheduling mechanism.

|  |  |  |  | GPT-2-1.5B ($M_t$) $\rightarrow$ GPT-2-0.1B ($M_s$) | | Qwen2.5-Math-7B-Inst ($M_t$) $\rightarrow$ Qwen2.5-Math-1.5B ($M_s$) | |
|---|---|---|---|---|---|---|---|
|  |  |  |  | **Avg.** | **P. Gains** | **Avg.** | **P. Gains** |
| KD Methods | (1) | (2) | (3) | *Rouge-L* | (%) | *pass@1* | (%) |
| KLD |  |  |  | 18.10 | - | 66.55 | - |
| +SpectrumKD (FS) | ✓ |  | ✓ | 19.07 | 5.34% | 67.90 | 2.03% |
| +SpectrumKD (FS) | ✓ | ✓ |  | 18.93 | 4.56% | 67.30 | 1.13% |
| +SpectrumKD (FS) |  |  | ✓ | 18.83 | 4.03% | 67.20 | 0.98% |
| +SpectrumKD (RS) | ✓ |  | ✓ | 18.45 | 1.91% | 66.60 | 0.08% |

Table 4: Component analysis of SpectrumKD's curriculum scheduler. Variants: with/without warm-up, forward (FS) vs. reverse (RS) sliding, linear vs. adaptive step size. Avg. performance gains (P. Gains) reported across tasks.

## 5.2 HYPERPARAMETER SENSITIVITY ANALYSIS

SpectrumKD introduces two key hyperparameters: the data partition ratios ($\lambda_a$, $\lambda_b$) and the distillation temperature range ($\tau_0$ to $\tau_n$). Figure 4 summarizes their sensitivity across model families and KD methods.

**Data Partition Ratios ($\lambda_a$ and $\lambda_b$).** As shown in Figure 4(a–b), performance is sensitive to both parameters but particularly to $\lambda_b$, which controls the size of the excluded (no-learning) zone. Since extreme hard instances can destabilize training, removing too few or too many samples degrades performance. An optimal, moderate partitioning—typically $\lambda_a = \lambda_b = 0.1$—maximizes distillation efficacy by balancing curriculum progression and data compatibility.

**Distillation Temperature ($\tau_0$ and $\tau_n$).** Figure 4(c–d) shows that a linearly increasing temperature schedule consistently outperforms fixed settings. Starting with $\tau_0 = 1$ enables the student to learn sharp, confident predictions early, while gradually raising $\tau$ to 2 allows it to absorb richer, softened teacher signals as its capacity grows—aligning supervision with learning dynamics.

### 5.3 COMPUTATIONAL COST ANALYSIS

SpectrumKD incurs overhead only during a one-time, offline preprocessing step: a single forward pass of the untrained student over the training set to compute cross-entropy losses, rank instances, and precompute epoch-specific subsets via the sliding window. No additional cost is added during training.

On the Dolly-15k dataset (11,435 samples) using OpenLLaMA2-3B, spectrum construction took 14 minutes and subset scheduling 2 minutes on four A100 80GB GPUs. Relative to total training time, this adds only $\sim$5.5% overhead for offline KD ($\sim$290 min) and $\sim$1.1% for online KD ($\sim$1500 min), confirming minor runtime impact.

### 5.4 COMPARISON WITH TRADITIONAL CURRICULUM LEARNING

We compare SpectrumKD against traditional curriculum learning (TCL) (Zhu et al., 2021), which uses handcrafted proxies (e.g., sentence-level entropy, sequence length) and applies a fixed schedule without filtering, as well as standard schedulers: Baby Step and One-Pass (Wang et al., 2021).

### 5.5 COMPARISON WITH TRADITIONAL CURRICULUM LEARNING

| | GPT-2-1.5B ($M_t$) $\rightarrow$ GPT-2-0.1B ($M_s$) | | OpenLLaMA2-7B ($M_t$) $\rightarrow$ OpenLLaMA2-3B ($M_s$) | |
|---|---|---|---|---|
| KD Methods | Avg. *Rouge-L* | P. Gains (%) | Avg. *Rouge-L* | P. Gains (%) |
| KLD | 18.10 | - | 22.56 | - |
| +SpectrumKD | 19.07 | 5.34% | 24.76 | 9.75% |
| +TCL (BS) | 18.28 | 0.99% | 22.91 | 1.55% |
| +TCL (OP) | 17.41 | -3.81% | 21.14 | -6.29% |

Table 5: Comparison with traditional curriculum learning (TCL) with standard schedulers (Baby Step (BS), One-Pass (OP)). Models: GPT-2 (1.5B$\rightarrow$0.1B) and OpenLLaMA2 (7B$\rightarrow$3B) with KLD. Avg. ROUGE-L and performance gains (P. Gains) reported. TCL uses entropy- and length-based difficulty without filtering.

As shown in Table 5, TCL offers marginal or even negative gains, while SpectrumKD delivers consistent improvements across models and tasks. This underscores the inadequacy of static, proxy-based difficulty metrics in LLM distillation and validates our distribution-aware, student-adaptive design. The results confirm that effective curriculum learning in KD requires both dynamic data curation and compatibility-aware scheduling—core principles embodied in SpectrumKD.

## 6 RELATED WORK

We discuss related work and defer a concentrated account to Appendix E.

## 7 CONCLUSION

We propose SpectrumKD, a plug-and-play framework for dynamic dataset curation in white-box KD of LLMs. Motivated by the critical role of data quality and student-model compatibility, SpectrumKD constructs a fixed difficulty spectrum from the global loss distribution of a training corpus and partitions it into four zones: Early Learning, Continuous Learning, Late Learning, and No Learning. This directly addresses the question of *what data to use* by filtering out incompatible extremes and prioritizing student-aligned instances. To answer *when to introduce data*, SpectrumKD employs a sliding window that progressively shifts from easier to harder regions of the spectrum, ensuring a smooth, curriculum-aligned increase in training difficulty that matches the student's evolving capacity. Crucially, SpectrumKD requires no changes to model architecture or distillation loss, integrating seamlessly into existing pipelines. Extensive experiments across instruction following, mathematical reasoning, and code generation—spanning multiple model families and KD objectives—demonstrate consistent performance gains with minor computational overhead. These results highlight SpectrumKD's generality, scalability, and practicality for real-world LLM compression. Future work includes developing task-adaptive difficulty metrics to further refine spectrum construction and enhance curation precision.

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

## A  PRELIMINARY FORMULATION OF WHITE-BOX KD

We present a formal characterization of white-box KD for auto-regressive language models, wherein a student model is trained not only on ground-truth target sequences but also on the soft probability distributions generated by a pre-trained teacher model. Formally, let the training corpus consist of prompt–response pairs $(x, y)$, where $x$ denotes the input prompt and $y = (y_1, \ldots, y_{|y|})$ represents the corresponding target token sequence. The student model, parameterized by $\theta$, is optimized with respect to a composite loss that integrates both supervised fine-tuning and knowledge distillation objectives.

The supervised fine-tuning component is formulated using the standard cross-entropy loss:

$$L_{ce} = -\sum_{i=1}^{|y|} \log q_\theta(y_i|x, y_{<i}), \tag{2}$$

where $q_\theta(y_i|x, y_{<i})$ denotes the conditional probability assigned by the student to the target token $y_i$, conditioned on the input prompt $x$ and the preceding tokens $y_{<i}$. This probability is derived via a softmax transformation applied to the student's raw logits $z^s$:

$$q_\theta(y_i|x, y_{<i}) = \frac{\exp(z^s_{y_i})}{\sum_{j \in V} \exp(z^s_j)}, \tag{3}$$

with $z^s_{y_i}$ indicating the logit corresponding to token $y_i$ and $V$ representing the full vocabulary.

Concurrently, the KD term encourages the student to emulate the teacher's per-token predictive distribution. The distillation loss is defined as:

$$L_{kd} = -\tau^2 \cdot \sum_{i=1}^{|y|} D\left(p(y_i|x, y_{<i}; \tau) \,\|\, q_\theta(y_i|x, y_{<i}; \tau)\right), \tag{4}$$

where $D(\cdot \,\|\, \cdot)$ denotes a statistical divergence between the temperature-scaled output distributions of the teacher $p$ and the student $q_\theta$. Prior work in white-box KD has explored a variety of divergence measures, including the standard Kullback–Leibler divergence (KLD) (Hinton et al., 2015), reverse

---

**Algorithm 1** SpectrumKD: Dynamic Dataset Curation for KD in LLM

---

**Input:** $D$: training dataset, $M_s$: student LM, $M_t$: teacher LM;
**Output:** $M_s^*$ : the distilled student LM.

1: CE-Loss = inference($M_s$,$D$);
2: $D_{ds}$ = sort($D$,CE-Loss);
3: $\lambda_a, \lambda_b$ = analyzer(CE-Loss);
4: $\{s_i | i \leq (1 - \lambda_b) \cdot N\}$ = $D'_{ds}$ = filter($D_{ds}, \lambda_a, \lambda_b$);
5: $D_{train} = \{s_i | i \leq (1 - \lambda_A - \lambda_b) \cdot N\}, \tau_0, n, k, t = 1$;
6: $M_s^* = M_s, ave.DL = \frac{max.DL}{n-k-1}$;
7: **for** $i = 1, ..., n$ **do**
8:    **if** $i \leq k + 1$ **then**
9:       $\tau = \tau_i$;
10:      $M_s^*$ = KD train($M_s^*$, $M_t$, $D_{train}$, $\tau$);
11:    **else**
12:      $\tau = \tau_i, \Delta = 0$;
13:      **while** $\Delta \leq ave.DL$ **do**
14:        $\Delta = \Delta + $ CE-Loss($s_{(1-\lambda_A-\lambda_b) \cdot N+t}$) $-$ CE-Loss($s_t$);
15:        $t = t + 1$;
16:        $D_{train} = \{s_i | t < i \leq (1 - \lambda_A - \lambda_b) \cdot N + t\}$
17:      **end while**
18:      $M_s^*$ = KD train($M_s^*$, $M_t$, $D_{train}$, $\tau$);
19:    **end if**
20: **end for**

---

KLD (Gu et al., 2023), Jensen–Shannon divergence (JSD) (Agarwal et al., 2024), total variation distance (TVD) (Wen et al., 2023), symmetrized KL (SKL) and symmetrized reverse KL (SRKL) (Ko et al., 2024), $\alpha$-$\beta$ divergence (Wang et al., 2025), task-aware information divergence (TAID) (Shing et al., 2025), and distribution-aware KD (DA-KD) (He et al.). The temperature parameter $\tau > 0$ modulates the sharpness of the output distributions, yielding softer probabilities that encapsulate richer relational information among tokens compared to hard one-hot labels.

The temperature-scaled distributions are computed as:

$$q_\theta(y_i | x, y_{<i}; \tau) = \frac{\exp(z_{y_i}^s / \tau)}{\sum_{j \in V} \exp(z_j^s / \tau)} \tag{5}$$

and

$$p(y_i | x, y_{<i}; \tau) = \frac{\exp(z_{y_i}^t / \tau)}{\sum_{j \in V} \exp(z_j^t / \tau)}, \tag{6}$$

where $z_j^t$ denotes the logits produced by the teacher model. The multiplicative factor $\tau^2$ in $L_{kd}$ is included to maintain consistent gradient magnitudes between $L_{ce}$ and $L_{kd}$ across different temperature settings, aligning with established conventions in the distillation literature (Hinton et al., 2015).

The total training objective is a convex combination of the two loss terms:

$$L_s = \alpha \cdot L_{ce} + (1 - \alpha) \cdot L_{kd}, \tag{7}$$

where $\alpha \in [0, 1]$ controls the relative emphasis placed on learning from ground-truth labels versus imitating the teacher's softened predictions.

## B  SPECTRUMKD TRAINING ALGORITHM

Algorithm 1 outlines the complete training pipeline of SpectrumKD, encompassing two core components: (1) the construction of a difficulty spectrum over the training dataset based on distributional divergence metrics, and (2) a sliding-window curriculum scheduling mechanism that dynamically selects training subsets according to evolving student proficiency.

## C  DETAILED EXPERIMENTAL SETUP

This section provides a thorough account of the experimental configuration, covering model architectures, datasets, training protocols, evaluation methodologies, and baseline approaches employed in our empirical study.

### C.1  BASE MODELS DESCRIPTION

Our experiments span three representative families of large language models—GPT-2, OpenL-LaMA2, and Qwen2.5—selected to reflect a diversity of architectures, training data regimes, and application domains. This selection ensures the generalizability of our findings across both general-purpose and domain-specialized settings. Specifically, GPT-2 and OpenLLaMA2 are utilized for general instruction-following tasks, while Qwen2.5 variants are deployed for mathematical reasoning and code generation. These models were chosen due to their open availability, architectural distinctiveness, and prevalence in established evaluation benchmarks, thereby supporting reproducibility and rigorous comparative analysis.

- **GPT-2** (Radford et al., 2019): A decoder-only Transformer architecture originally developed by OpenAI. Within our knowledge distillation framework, we adopt GPT-2 (1.5B parameters) as the teacher and GPT-2 (0.1B parameters) as the student, both fine-tuned for instruction-following. This intra-family pairing—characterized by a substantial capacity gap—enables a controlled investigation into how knowledge transfers across scales within a consistent architectural paradigm.

- **OpenLLaMA2** (Geng & Liu, 2023): An open-source, permissively licensed reimplementation of Meta's LLaMA model, trained exclusively on the RedPajama dataset (Computer, 2023) using publicly accessible data, in contrast to the proprietary corpus of the original LLaMA (Touvron et al., 2023). We instantiate OpenLLaMA2 (7B) as the teacher and OpenLLaMA2 (3B) as the student for instruction-following tasks. This setup allows us to assess distillation performance in models developed under open, community-driven paradigms with transparent data provenance.

- **Qwen2.5** (Team, 2024): A family of domain-specialized large language models released by the Qwen team, optimized for high-performance reasoning in specific technical domains. For mathematical reasoning, we employ Qwen2.5-Math-7B-Inst as the teacher and Qwen2.5-Math-1.5B as the student; for code generation, we use Qwen2.5-Coder-7B-Inst as the teacher and Qwen2.5-Coder-1.5B as the student. These targeted pairings facilitate a fine-grained evaluation of distillation efficacy in vertically integrated, task-optimized architectures.

All models are initialized from publicly released checkpoints, and all fine-tuning procedures adhere to identical optimization hyperparameters and training protocols to ensure a fair and controlled comparison across distillation settings.

### C.2  DATASET DESCRIPTION

We evaluate the SpectrumKD framework across three representative natural language processing domains: instruction-following, mathematical reasoning, and code generation. Below, we detail the datasets employed for training and evaluation in each task category.

- `databricks-dolly-15k` (instruction-following; (Conover et al., 2023)): A human-curated dataset comprising 15,000 instruction–response pairs spanning seven categories, such as brainstorming, closed-ended question answering, and summarization. Its high-quality, naturally phrased prompts make it well-suited for training and assessing instruction-following capabilities.

- `self-instruct-eval` (instruction-following; (Wang et al., 2022a)): A synthetic instruction dataset generated via self-instruct bootstrapping. We utilize its training split (52,000 instructions yielding 82,000 input–output pairs) and its evaluation set (252 expert-defined tasks with 50,000 public examples) to probe generalization to novel task formulations.

- `vicuna-eval` (instruction-following; (Chiang et al., 2023)): A benchmark of 80 complex, open-ended prompts requiring multi-step reasoning and nuanced comprehension. Widely adopted for evaluating conversational agents, it serves as a stringent test of instruction adherence and depth of reasoning.

- `supernatural-instructions` (instruction-following; (Wang et al., 2022b)): A large-scale benchmark encompassing 1,616 expert-authored NLP tasks across 76 task types. Its test set includes approximately 9,000 samples from 119 held-out tasks, enabling robust evaluation of zero-shot generalization.

- `unnatural-instructions-core` (instruction-following; (Honovich et al., 2022)): A synthetically constructed dataset of 66,000 instruction–response pairs generated by automatically perturbing natural instructions. The core subset is used to study model behavior when trained predominantly on machine-generated data.

- `MetaMathQA` (mathematical reasoning; (Yu et al., 2023)): A dataset of 39,500 math problems generated through iterative question bootstrapping involving forward and backward reasoning transformations. Designed to improve generalization, it emphasizes diverse problem-solving strategies.

- `GSM8K` (mathematical reasoning; (Cobbe et al., 2021)): A collection of 8,500 grade-school-level math word problems requiring multi-step arithmetic reasoning. Each problem necessitates explicit chain-of-thought decomposition, making it a standard benchmark for evaluating reasoning fidelity in distilled models.

- `MATH` (mathematical reasoning; (Hendrycks et al., 2021)): A dataset of competition-style mathematical problems covering algebra, geometry, number theory, and other domains at varying difficulty levels. It is designed to assess deep mathematical understanding and symbolic reasoning capabilities.

- `Evol-Instruct-Code-80k-v1` (code generation; (Luo et al., 2023)): Constructed via the Evol-Instruct methodology, this dataset begins with Code Alpaca (20K samples) and iteratively applies instruction evolution techniques—such as constraint injection, reasoning depth amplification, misleading code insertion, and complexity requirements—to produce 78K high-quality, progressively challenging code instructions. It has been shown to significantly enhance code generation performance when used for fine-tuning.

- `MBPP` (code generation; (Austin et al., 2021)): A crowd-sourced dataset of 974 programming problems, each consisting of a natural language description (docstring), a function signature, and three test cases. A verified subset ensures high solution correctness, making it suitable for reliable evaluation.

- `HumanEval` (code generation; (Chen et al., 2021)): A collection of 164 handcrafted programming tasks, each providing a function signature and docstring as input. These problems were explicitly designed to be absent from common training corpora, ensuring an unbiased assessment of code synthesis capabilities.

## C.3 TRAINING DETAILS

All experiments are conducted on a uniform hardware setup comprising four NVIDIA A100 80GB GPUs and an Intel(R) Xeon(R) Platinum 8350C CPU to ensure consistency across training runs.

**Hyperparameter Settings.** We adopt a consistent set of hyperparameters for the SpectrumKD framework across all tasks. The spectral regularization coefficients are set to $\lambda_a = 0.1$ and $\lambda_b = 0.1$. The distillation temperature $\tau$ is scheduled to increase linearly from an initial value of $\tau_0 = 1$ to a final value of $\tau_n = 2$. For off-policy knowledge distillation, the supervised fine-tuning (SFT) loss weight is initialized at $\alpha_0 = 0.3$. In contrast, on-policy distillation omits the SFT term entirely ($\alpha = 0$ throughout training), as the student learns solely from its own generations refined via teacher feedback. Additionally, we implement the optimal mixing strategy proposed by Agarwal et al. (2024), combining 50% self-generated outputs (SGOs) with 50% ground-truth responses to stabilize training dynamics and improve generalization.

Model checkpoint selection is task-dependent: for instruction-following, we select the best checkpoint based on Rouge-L scores on the validation set, which have been shown to correlate strongly

with human judgments in summarization and instruction tasks (Agarwal et al., 2024); for mathematical reasoning and code generation, we use *pass@1* on the validation set as the selection criterion.

**Instruction-Following Experiments.** We use the `databricks-dolly-15k` dataset (Conover et al., 2023), randomly split into 12.5K training, 1K validation, and 0.5K test instances. Samples exceeding the model's maximum context length are filtered prior to training. Both teacher models—GPT-2 (1.5B) and OpenLLaMA2 (7B)—are first fine-tuned on this dataset before distillation.

Hyperparameter tuning is performed via grid search over learning rates {5e-4, 1e-4, 5e-5} and batch sizes {8, 16}, guided by validation performance. Baseline models are trained for 20 epochs. In contrast, SpectrumKD uses only 80% of the baseline dataset per epoch; to maintain parity in total optimization steps, it is trained for 25 epochs, with the first 5 epochs designated as a warm-up phase.

For OpenLLaMA2, we employ Low-Rank Adaptation (LoRA) (Hu et al., 2022), a parameter-efficient fine-tuning technique. All linear layers in the self-attention and MLP subnetworks of the Transformer architecture are designated as LoRA target modules.

**Mathematical Reasoning Experiments.** We sample 20K training and 2K validation instances from the `MetaMathQA` dataset (Yu et al., 2023). To encourage structured reasoning, we prepend each prompt with the instruction: *"Please reason step by step, and put your final answer within* `\boxed{}`*."* following the chain-of-thought prompting paradigm (Wei et al., 2022).

Baseline models are trained for 4 epochs with a fixed learning rate of $5 \times 10^{-5}$. SpectrumKD, using 80% of the baseline data per epoch, is trained for 5 epochs with a 2-epoch warm-up phase. Across all configurations, we maximize per-GPU batch size under memory constraints and use gradient accumulation to maintain an effective batch size of 128. LoRA (Hu et al., 2022) is applied with the same target modules as in the instruction-following experiments.

**Code Generation Experiments.** We draw 20K training and 2K validation samples from the `Evol-Instruct-Code-80k-v1` dataset (Luo et al., 2023). Baseline models are trained for 4 epochs at a learning rate of $5 \times 10^{-5}$. SpectrumKD follows the same data reduction strategy (80% per epoch) and is trained for 5 epochs with a 2-epoch warm-up. Gradient accumulation is again used to achieve an effective batch size of 128 across 4 A100 GPUs. LoRA (Hu et al., 2022) is consistently applied to all self-attention and MLP linear layers to ensure parameter efficiency and training stability.

## C.4 EVALUATION DETAILS

All evaluations are conducted on a single NVIDIA A100 80GB GPU, adhering to established protocols from prior work: we follow the evaluation setup of Gu et al. (2023) for instruction-following experiments, that of Yang et al. (2024b) for mathematical reasoning, and the methodology of Hui et al. (2024) for code generation tasks.

**Instruction-Following Experiments.** To ensure consistent and unbiased inference, we apply a fixed prompt template (Figure 5) across all models. Responses are generated with a temperature of 1.0 and a maximum output length of 512 tokens. To mitigate stochastic variance and enhance result reliability, we produce five independent responses per input using distinct random seeds ({10, 20, 30, 40, 50}) and report metrics averaged over these replicates.

For LLM-as-a-Judge evaluation (Zheng et al., 2023), we employ the pairwise comparison prompt shown in Figure 6, configured with a temperature of 0.7. Within the GPT-2 model family, student outputs are compared against those of the vanilla GPT-2 (1.5B) teacher; for the OpenLLaMA2 family, comparisons are made relative to the vanilla OpenLLaMA2 (7B) teacher.

**Mathematical Reasoning & Code Generation Experiments.** We utilize task-specific prompt templates during inference: Figure 7 for mathematical reasoning and Figure 8 for code generation. Following Yang et al. (2024b), we adopt an 8-shot prompt for GSM8K and a 4-shot prompt for MATH to elicit chain-of-thought reasoning, with all in-context examples sourced from their work. Generation is performed using greedy decoding with a maximum output length of 1024 tokens to accommodate extended reasoning traces. For code generation, we evaluate model outputs using the EvalPlus framework (Liu et al., 2023), which provides enhanced test suites for robust correctness assessment.

```
Below is an insruction that describes a task.
Write a response that appropriately completes the request.

### Instruction:
{instruction}

### Input:
{input}

### Response:
```

Figure 5: Prompt template used in instruction-following experiments, adapted from (Gu et al., 2023)

```
### System:
Please act as an impartial judge and evaluate the quality of
the responses provided by two AI assistants to the user
question displayed below. You should choose the assistant
that follows the user's instructions and answers the user's
question better. Your evaluation should consider factors such
as the helpfulness, relevance, accuracy, depth, creativity,
and level of detail of their responses. Begin your evaluation
by comparing the two responses and provide a short
explanation. Avoid any position biases and ensure that the
order in which the responses were presented does not
influence your decision. Do not allow the length of the
responses to influence your evaluation. Do not favor certain
names of the assistants. Be as objective as possible. After
providing your explanation, output your final verdict by
strictly following this format: "[[A]]" if assistant A is
better, "[[B]]" if assistant B is better, and "[[C]]" for
a tie.

### User Question:
{question}

[The Start of Assistant A's Answer]
{answer a}
[The End of Assistant A's Answer]

[The Start of Assistant B's Answer]
{answer b}
[The End of Assistant B's Answer]
```

Figure 6: The pairwise comparison prompt introduced in LLM-as-a-Judge, adapted from (Zheng et al., 2023)

```
<8 examples> # for GSM8K
<4 examples> # for MATH

### Problem:
{instruction}
Please reason step by step, and put your final answer within
\boxed{}.

### Solution:
```

Figure 7: The prompt used in mathematical reasoning experiments, adapted from (Yang et al., 2024b)

```
Please provide a self-contained Python script that solves the
following problem in a markdown code block:

### Problem:
{instruction}

### Response:
Below is a Python script with a self-contained function that
solves the problem and passes corresponding tests:
```

Figure 8: The prompt used in code generation experiments, adapted from (Hui et al., 2024)

### C.5  BASELINE DESCRIPTION

This section describes the knowledge distillation (KD) loss functions included in our comparative study. Each baseline quantifies the discrepancy between the teacher's output distribution $p$ and the student's distribution $q_\theta$ using a distinct statistical divergence measure. To ensure a thorough and representative comparison, we incorporate both widely adopted and recently proposed divergence-based objectives from the distillation literature.

The KLD (Hinton et al., 2015), a standard choice in KD, is defined as:

$$D_{KLD}(p \parallel q_\theta) = \mathbb{E}_{y \sim p} \left[ \log \frac{p(y)}{q_\theta(y)} \right],$$ (8)

where the expectation is taken over token predictions drawn from the teacher's distribution $p$. This formulation encourages the student to assign high probability to outputs favored by the teacher.

The Reverse KLD (RKL), which reverses the argument order, is given by:

$$D_{RKL}(q_\theta \parallel p) = \mathbb{E}_{y \sim p} \left[ \log \frac{q_\theta(y)}{p(y)} \right].$$ (9)

Unlike KLD, RKL emphasizes fitting the student distribution to the teacher's high-probability regions and has been shown to improve stability in certain distillation settings.

The Jensen–Shannon Divergence (JSD) (Agarwal et al., 2024) provides a symmetric and smoothed measure of divergence:

$$D_{JSD}(p, q_\theta) = \frac{1}{2} \mathbb{E}_{y \sim p} \left[ \log \frac{p(y)}{m(y)} \right] + \frac{1}{2} \mathbb{E}_{y \sim q_\theta} \left[ \log \frac{q_\theta(y)}{m(y)} \right],$$ (10)

where $m(y) = \frac{1}{2}p(y) + \frac{1}{2}q_\theta(y)$ is the midpoint distribution. JSD balances bidirectional alignment and is less sensitive to distribution mismatches than asymmetric divergences.

The Total Variation Distance (TVD) (Wen et al., 2023) measures the $\ell_1$ difference between the two distributions:

$$D_{TVD}(p, q_\theta) = \frac{1}{2} \sum_y |p(y) - q_\theta(y)|.$$ (11)

As a metric bounded in $[0, 1]$, TVD offers intuitive interpretability and robustness to outliers, though it lacks gradient signals in regions of non-overlapping support.

The Forward Skew KLD (SKL) (Ko et al., 2024) introduces a convex combination of teacher and student distributions in the reference argument:

$$D_{SKL}(p \parallel q_\theta) = D_{KLD} \left( p \parallel \beta p + (1 - \beta) q_\theta \right),$$ (12)

while the Reverse Skew KLD (SRKL) is defined as:

$$D_{SRKL}(q_\theta \parallel p) = D_{KLD}\left(q_\theta \parallel (1-\beta)p + \beta q_\theta\right), \tag{13}$$

where $\beta \in [0, 1]$ controls the interpolation weight. These skewed variants mitigate extreme gradients by smoothing the target distribution, enhancing training stability.

Collectively, these divergence functions represent a diverse set of strategies for aligning student and teacher models in white-box KD, covering symmetric, asymmetric, and regularized forms of distribution matching. Their inclusion demonstrates the flexibility of the proposed SpectrumKD framework in accommodating diverse KD loss functions, while consistently achieving strong performance across different divergence measures, underscoring its robustness and adaptability in practical distillation scenarios.

## D   ADDITIONAL EXPERIMENTAL RESULTS

This section presents supplementary empirical findings that further validate the efficacy and robustness of the SpectrumKD framework.

Table 6 reports evaluation results for GPT-2 as the student model, including ROUGE-L scores and pairwise winning rates averaged across five random seeds. These results demonstrate the consistent performance gains and generalizability of SpectrumKD across different large language model architectures under knowledge distillation.

| | GPT-2-1.5B ($M_t$) $\rightarrow$ GPT-2-0.1B ($M_s$) | | | | | | | | | | | | | |
| | DollyEval | | SelfInst | | VicunaEval | | S-NI | | UnNI | | Avg. | | P. Gains | |
| KD Methods | R-L | WR | R-L | WR | R-L | WR | R-L | WR | R-L | WR | R-L | WR | (%) | (%) |
|---|---|---|---|---|---|---|---|---|---|---|---|---|---|---|
| $M_t$ | 27.43 | 52.79 | 14.12 | 52.15 | 16.37 | 52.85 | 27.68 | 53.44 | 31.92 | 52.95 | 23.50 | 52.84 | - | - |
| SFT | 23.33 | 47.04 | 10.01 | 42.28 | 14.72 | 48.31 | 16.38 | 36.32 | 19.57 | 39.37 | 16.80 | 42.67 | - | - |
| GKD | 24.67 | 48.57 | 11.48 | 45.51 | 15.66 | 49.94 | 23.81 | 45.60 | 25.26 | 45.42 | 20.18 | 47.01 | - | - |
| +SpectrumKD | 24.61 | 49.65 | 11.81 | 47.37 | 15.72 | 51.12 | 26.38 | 48.75 | 27.68 | 48.77 | 21.24 | 49.13 | 5.27 | 4.51 |
| KLD | 23.49 | 47.30 | 10.33 | 42.78 | 14.96 | 49.11 | 19.71 | 40.80 | 22.01 | 42.04 | 18.10 | 44.41 | - | - |
| +SpectrumKD | 24.71 | 49.34 | 11.31 | 45.85 | 15.21 | 50.09 | 20.84 | 43.19 | 23.26 | 44.40 | 19.07 | 46.58 | 5.34 | 4.88 |
| RKL | 23.79 | 47.87 | 12.13 | 46.96 | 14.94 | 48.67 | 23.81 | 45.38 | 22.52 | 42.63 | 19.44 | 46.30 | - | - |
| +SpectrumKD | 24.53 | 49.03 | 12.14 | 48.04 | 15.01 | 50.12 | 24.32 | 46.89 | 24.41 | 45.62 | 20.08 | 47.94 | 3.31 | 3.53 |
| JSD | 24.07 | 48.66 | 11.38 | 45.59 | 15.87 | 50.79 | 22.84 | 44.20 | 23.06 | 43.30 | 19.44 | 46.51 | - | - |
| +SpectrumKD | 24.82 | 49.94 | 11.12 | 45.56 | **16.01** | **51.93** | 26.14 | 48.91 | 24.53 | 45.60 | 20.52 | 48.39 | 5.55 | 4.04 |
| TVD | 24.32 | 48.25 | 11.09 | 45.01 | 15.51 | 49.70 | 25.93 | 47.66 | 26.55 | 46.96 | 20.68 | 47.52 | - | - |
| +SpectrumKD | 24.63 | 49.75 | 12.48 | 48.47 | 15.45 | 51.45 | **28.03** | **50.65** | 28.74 | 49.80 | 21.97 | 50.02 | 6.22 | 5.28 |
| SKL | 24.24 | 48.64 | 12.27 | 47.23 | 15.71 | 50.15 | 23.33 | 45.06 | 24.02 | 44.25 | 19.91 | 47.07 | - | - |
| +SpectrumKD | 25.38 | 50.50 | 12.85 | 49.09 | 15.81 | 51.27 | 26.31 | 48.78 | 27.84 | 48.97 | 21.64 | 49.72 | 8.66 | 5.64 |
| SRKL | 25.22 | 49.13 | 12.86 | 48.21 | 15.18 | 49.18 | 25.51 | 47.18 | 28.43 | 48.59 | 21.44 | 48.46 | - | - |
| +SpectrumKD | **25.68** | **50.58** | **13.23** | **49.91** | 15.95 | 51.73 | 27.94 | 49.75 | **29.32** | **50.13** | **22.42** | **50.42** | 4.59 | 4.05 |

Table 6: Instruction-following evaluation on GPT-2. $M_t$ and $M_s$ denote teacher and student models. Baselines include SFT and seven white-box KD methods: GKD (on-policy, using 50% student generated outputs), and off-policy variants (KLD, RKL, JSD, TVD, SKL, SRKL). **DollyEval**, **SelfInst**, **VicunaEval**, **S-NI**, and **UnNI** report Rouge-L scores (R-L) and winning rates (WR); **Avg.** is the mean across datasets. Best student results are **bold**; Results are averaged over five random seeds.

Table 7 provides an ablation study examining various metrics used to quantify instance difficulty during the construction of the difficulty spectrum. This analysis highlights the sensitivity of distillation performance to the choice of difficulty estimator and underscores the rationale behind our selected metric.

| | GPT-2-1.5B ($M_t$) $\rightarrow$ GPT-2-0.1B ($M_s$) | | | | | Qwen2.5-Math-7B-Inst ($M_t$) $\rightarrow$ Qwen2.5-Math-1.5B ($M_s$) | |
| | DollyEval | SelfInst | VicunaEval | S-NI | UnNI | GSM8K | MATH |
| KD Methods | Rouge-L | Rouge-L | Rouge-L | Rouge-L | Rouge-L | pass@1 | pass@1 |
|---|---|---|---|---|---|---|---|
| KLD | 23.49 | 10.33 | 14.96 | 19.71 | 22.01 | 77.9 | 55.2 |
| +SpectrumKD (CE-LOSS) | 24.71 | 11.31 | 15.21 | 20.84 | 23.26 | 78.7 | 57.1 |
| +SpectrumKD (ROUGE-L) | 24.73 | 11.28 | 15.25 | 20.75 | 23.31 | 78.0 | 55.6 |
| +SpectrumKD (SE) | 24.02 | 10.81 | 14.85 | 19.93 | 22.13 | 78.3 | 56.2 |

Table 7: Ablation on difficulty metrics for spectrum construction: CE-LOSS (cross-entropy), ROUGE-L, and SE (sentence entropy). Avg. performance gains (P. Gains) over baseline KD (KLD) shown for GPT-2 and Qwen2.5-Math.

Table 8 investigates the contribution of key components in the SpectrumKD pipeline: (1) forward sliding curriculum (i.e., progressive selection of instances from easy to hard), (2) inclusion of a warm-up training stage, and (3) adaptive adjustment of the sliding window step size. The results illustrate the individual and collective impact of these design choices on final model performance.

| | | | | GPT-2-1.5B ($M_t$) $\rightarrow$ GPT-2-0.1B ($M_s$) | | | | | Qwen2.5-Math-7B-Inst ($M_t$) $\rightarrow$ Qwen2.5-Math-1.5B ($M_s$) | |
| | | | | **DollyEval** | **SelfInst** | **VicunaEval** | **S-NI** | **UnNI** | **GSM8K** | **MATH** |
| KD Methods | (1) | (2) | (3) | *Rouge-L* | *Rouge-L* | *Rouge-L* | *Rouge-L* | *Rouge-L* | *pass@1* | *pass@1* |
|---|---|---|---|---|---|---|---|---|---|---|
| KLD | | | | 23.49 | 10.33 | 14.96 | 19.71 | 22.01 | 77.9 | 55.2 |
| +SpectrumKD (FS) | √ | | √ | 24.71 | 11.31 | 15.21 | 20.84 | 23.26 | 78.7 | 57.1 |
| +SpectrumKD (FS) | √ | √ | | 24.68 | 11.24 | 15.12 | 20.54 | 23.05 | 78.5 | 56.1 |
| +SpectrumKD (FS) | | | √ | 24.34 | 11.14 | 15.06 | 20.46 | 23.15 | 78.6 | 55.8 |
| +SpectrumKD (RS) | √ | | √ | 23.61 | 10.78 | 15.04 | 19.89 | 22.91 | 77.8 | 55.4 |

Table 8: Component analysis of SpectrumKD's curriculum scheduler. Variants: with/without warm-up, forward (FS) vs. reverse (RS) sliding, linear vs. adaptive step size. Avg. P. Gains reported across tasks.

# E   RELATED WORK

**White-Box Knowledge Distillation for LLMs.** White-box KD for LLMs leverages access to the teacher's internal logits or soft probability distributions, enabling more effective transfer than black-box approaches (Yang et al., 2024c). A significant body of work in this setting has focused on designing divergence measures for the distillation loss. While the Kullback–Leibler divergence (KLD) remains a standard choice (Hinton et al., 2015), its asymmetry can induce mode-averaging behavior that degrades distillation quality (Gu et al., 2023). To address this, recent studies have proposed alternatives including reverse KLD (RKL) (Gu et al., 2023), Jensen–Shannon divergence (JSD) (Agarwal et al., 2024), skew KLD variants (SKL and SRKL) (Ko et al., 2024), $\alpha$-$\beta$-Divergence (Wang et al., 2025), and task-aware information divergence (TAID) (Shing et al., 2025). Despite their success in specific contexts, these divergence-based objectives often exhibit inconsistent generalization across tasks and datasets (Agarwal et al., 2024; Ko et al., 2024), underscoring the limitations of loss-centric approaches and motivating the need for complementary data-centric strategies.

**Data Curation in Knowledge Distillation.** Complementary to loss design, data curation strategies—such as incorporating teacher generated outputs or student generated outputs—have been shown to improve training stability and performance (Kim & Rush, 2016; Agarwal et al., 2024). However, these methods frequently neglect the importance of data quality and alignment with the student's learning capacity, principles well-supported in both SFT and black-box KD literature (Li et al., 2023a; 2024; Ding et al., 2023). For example, UltraChat (Ding et al., 2023) emphasizes high-quality, diverse data curation for black-box distillation, while the *phi* series (Gunasekar et al., 2023) demonstrates the efficacy of distilling from compact, expert-authored corpora. More recently, Liu & Zhang (2025b) introduced a white-box KD framework that refines training data through iterative student-driven reflection, highlighting the emerging recognition of data quality as a critical distillation dimension.

**Curriculum Learning in Knowledge Distillation.** Curriculum learning (CL) has been explored to govern the sequencing of training data, though its application has been largely confined to computer vision (Xiang et al., 2020; Li et al., 2023b) with limited adoption in NLP (Zhu et al., 2021). Recent KD-specific CL methods include MPDistil (Sengupta et al., 2023), which employs reinforcement learning for curriculum scheduling; Confucius (Gao et al., 2024), designed for tool-augmented learning via black-box KD; POCL (Liu & Zhang, 2025a), a plug-and-play framework inspired by progressive overload principles; and DA-KD (He et al.), which dynamically adjusts the distillation dataset based on sample difficulty. Nevertheless, these approaches generally fail to jointly account for the interplay among instance difficulty, data fidelity, and the evolving capacity of the student model—a key gap that SpectrumKD aims to address in the white-box KD setting.

## F  LIMITATIONS

SpectrumKD is grounded in the empirical observation that the distribution of instance difficulty—measured via cross-entropy loss—often follows a log-normal pattern, characterized by a sharp peak and a long right tail. This implies that extremely hard instances are inherently present in most datasets, albeit with rapidly diminishing frequency. However, this distributional assumption is sensitive to both the choice of difficulty metric and the intrinsic properties of the underlying dataset. Our current study offers limited investigation into the behavior of SpectrumKD when the difficulty spectrum deviates significantly from log-normality.

Furthermore, SpectrumKD relies on token-level cross-entropy loss to construct the difficulty spectrum. While effective for instruction-following tasks, this metric yields more modest gains in mathematical reasoning and code generation. We hypothesize that this stems from the fact that correctness in these domains is highly sensitive to small, discrete errors—such as arithmetic miscalculations or syntactic bugs—that are poorly captured by token-wise likelihood. Consequently, cross-entropy may assign high scores to outputs that superficially resemble reference solutions but are functionally incorrect. This limitation points to the need for task-aware difficulty estimation mechanisms in KD. Future work should explore integrating execution-based feedback—such as code compilation results, test case execution outcomes, or mathematical answer verification—into the dynamic data curation pipeline to better align difficulty assessment with functional correctness.

## G  STATEMENT ON LLM USAGE

We disclose that large language models were used to assist in the preparation of this manuscript. Specifically, an LLM was employed to (1) identify and correct grammatical errors and (2) refine phrasing to align with conventions of academic English writing. Additionally, LLMs were utilized during the implementation phase to generate boilerplate code, verify logic, and debug scripts, ensuring the correctness and reproducibility of our experimental pipeline. All scientific content, experimental design, analysis, and interpretation remain the sole responsibility of the authors.

