# OpenReview forum: "SpectrumKD: Dynamic Dataset Curation for Distribution-Aware Knowledge Distillation of Large Language Models"
_ICLR.cc/2026/Conference — ICLR 2026 Conference Withdrawn Submission_

### Official Review · Reviewer_Trz4 · 2025-10-25

**Soundness:** 2
**Presentation:** 3
**Contribution:** 3
**Rating:** 2
**Confidence:** 4

**Summary:**

SpectrumKD is a pragmatic data-curation layer for white-box knowledge distillation. The authors first compute per-example cross-entropy with an untrained student to build a global “difficulty spectrum,” partitioned into four zones (Early/Continuous/Late/No Learning). Training then uses a fixed-size sliding window that moves from easy to hard across epochs so that the aggregate difficulty of the active subset increases roughly uniformly. A linear temperature ramp softens teacher logits in sync with this progression. The module is drop‑in—it does not change model architectures or KD losses—and is evaluated with several objectives (KLD, JSD, SRKL, on‑policy GKD) across instruction following, math reasoning, and code generation. Empirically, SpectrumKD delivers consistent, modest gains with small overhead, and the paper includes ablations on difficulty metrics, zone thresholds, and scheduling choices.

**Strengths:**

- Practical and plug‑and‑play: one offline scoring pass and a lightweight scheduler; compatible with many KD losses and model pairs.
- Clear, coherent design: difficulty spectrum → four-zone partition → sliding window → temperature ramp; figures make the workflow easy to follow.
- Broad empirical sweep: multiple tasks, families (GPT‑2/OpenLLaMA2/Qwen2.5), and losses; component ablations and sensitivity studies are provided.
- Sensible motivation: emphasizes dataset distribution and student–data compatibility rather than only instance‑level “informativeness,” which is often overlooked in KD.
- Low engineering cost: improvements achieved without touching core KD objectives or model code; overhead appears minor.

**Weaknesses:**

- Limited conceptual novelty: CE‑based difficulty ranking, curriculum‑style progression, and temperature ramping are established ideas; the contribution reads as a careful integration rather than a new principle. The distinction from competence‑based CL, uncertainty/divergence sampling (e.g., teacher–student KL, SKD/DDS), and budgeted on‑policy data selection is not sufficiently sharp.
- Confounded difficulty metric: cross‑entropy correlates with sequence length, domain, and templating. The paper does not report partial‑correlation or length‑controlled analyses, nor task‑specific difficulty signals (e.g., executability for code, logical step correctness for math). This leaves open whether the spectrum primarily captures length/style rather than genuine hardness.
- Budget fairness is under-specified: filtering out extreme hard samples may change effective update density. It’s unclear whether baselines are matched on tokens, steps, and wall‑clock time; stronger recent sampling baselines under strict budget parity are missing.
- Generalization gaps: results focus on white‑box KD; applicability to black‑box settings, larger modern teachers (e.g., Llama‑3/Mixtral), and stronger students (≥7B) is not demonstrated.
- Reporting and reproducibility: significance testing is inconsistent across tables; implementation details are scattered. Releasing scoring caches, subset indices, and configs would materially improve reproducibility.

**Questions:**

1) Positioning and novelty
- In one sentence, what is the genuinely new principle beyond integrating CE-based difficulty + curriculum sliding + temperature ramp? What can SpectrumKD do that competence-based CL or divergence/uncertainty sampling cannot?
- Which prior methods is SpectrumKD most likely to be confused with? Please spell out the decisive differences and why those matter empirically.

2) Difficulty metric and confounds
- How correlated is your CE-based difficulty with sequence length and domain/source? Could you share a simple correlation table or length-bucket analysis to show the spectrum isn’t just a length proxy?
- For math/code, CE can miss “one critical mistake” semantics. Have you tried task-specific difficulty signals (e.g., executability for code, step-consistency for math)? Do they change the spectrum or results meaningfully?
- What happens with noisy or mislabeled data—does the method simply banish them to “No Learning,” and could that hurt robustness?

3) Budget fairness
- Across all main tables, are tokens, steps, and wall-clock strictly matched to baselines? If not all three, which two are matched, and where could mismatches inflate gains?
- Filtering out extreme hard samples can increase effective update density. Can you run a control that keeps the full dataset but adjusts steps/learning rate so “effective updates” are comparable?

4) Scheduler behavior
- Please describe, in concrete terms, how the window moves when the spectrum has cliffs or multiple modes. How do you prevent oscillation or big jumps? Any smoothing or max-step constraints?

5) Temperature vs. difficulty
- Are the gains from the temperature ramp independent of the sliding window? A small 2D ablation (several ramps × several sliding schemes) would clarify whether one carries most of the lift.

6) Thresholds and adaptivity
- Are the four-zone cutoffs fixed across tasks, or tuned? Would a simple adaptive rule (e.g., quantiles chosen to stabilize validation loss) work as well or better?

7) Treatment of very hard examples
- Instead of excluding them forever, did you try bringing them back late with a small weight (hard replay/contrastive replay)? Any effect on robustness or long‑tail generalization?

8) Scope and scalability
- Do you expect similar benefits in black-box KD (teacher logits only)? What’s the simplest way to approximate your spectrum there?
- Have you tried larger modern teachers (e.g., Llama‑3/Mixtral) or stronger students (≥7B)? Do optimal window sizes/thresholds shift with scale?

9) Metrics and significance
- Can you report variance and significance consistently across the main results, and—where feasible—add stronger task-relevant metrics (e.g., executable pass@k for code, better LLM-as-a-judge or limited human eval for instruction)?

10) Reproducibility and data transparency
- Will you release the scoring cache, subset indices, and exact configs to let others reproduce the curves? Also, a brief note on dataset licensing/filters and any basic bias checks would be helpful.

---

### Official Review · Reviewer_Q38k · 2025-10-31

**Soundness:** 3
**Presentation:** 3
**Contribution:** 2
**Rating:** 4
**Confidence:** 3

**Summary:**

The paper introduces SpectrumKD, a dynamic dataset curation framework designed to enhance knowledge distillation for large language models by prioritizing global distribution-aware data selection. SpectrumKD constructs a difficulty spectrum by ranking all training instances based on their cross-entropy loss as evaluated by an initial student model. This spectrum is then partitioned into four distinct zones. Based on this partitioning, the framework employs a sliding window curriculum scheduler that progressively shifts across the spectrum from easier to harder instances over the course of training epochs. The method is evaluated across multiple benchmarks, demonstrating consistent performance gains.

**Strengths:**

The partitioning of data into distinct learning phases based on difficulty, coupled with adaptive curriculum scheduling, is well-motivated by empirical and theoretical insights. The plug-and-play design, which integrates seamlessly with existing KD methods without modifying core algorithms, is a practical strength.

**Weaknesses:**

1. As the authors mention in the limitations, SpectrumKD is primarily based on the assumption that the distribution of instance difficulty follows a log-normal pattern, which depends on the distribution of the dataset. For datasets whose difficulty spectrum deviates significantly from log-normality (e.g., those that are extremely easy or extremely difficult), the values of λa and λb may be severely skewed, and the effectiveness of SpectrumKD has not been validated in such cases.
2. In Section 3.2, the difficulty metric is defined as the cross-entropy loss Li = −log qθ(yi|xi). It is unclear whether this refers to the total sequence loss or the length-normalized (average) loss. Using the total loss would introduce a significant bias, conflating sequence length with intrinsic difficulty.
3. There are several presentation issues that merit correction: duplicate section titles (“Comparison with Traditional Curriculum Learning” appears twice in Section 5.4 and Section 5.5), minor article/capitalization errors (e.g., “an uniform” should be “a uniform”), and inconsistent table cross-references (e.g., line 320 referring to “Table 3” for main instruction-following results, which are in Table 1 here).

**Questions:**

1. Since the learning ability of the student model increases during training, the initial estimation of instance difficulty may be biased. Have you tried periodically re-estimating the difficulty spectrum (e.g., every few epochs) to adapt to the evolving student model, thereby enabling a more dynamic approach to dataset curation?
2. Is the performance where curriculum scheduler (1), (2), and (3) are used simultaneously missing in Table 4?

---

### Official Review · Reviewer_UAZg · 2025-10-31

**Soundness:** 2
**Presentation:** 2
**Contribution:** 2
**Rating:** 2
**Confidence:** 4

**Summary:**

This paper proposes SpectrumKD, a curriculum learning-based data utilization strategy for traditional off-policy distillation. The method measures the difficulty of data based on the cross-entropy loss of the untrained student model, using thresholding to categorize instance difficulty. The most difficult fraction of samples are discarded during training, while the remaining samples are learned in a progressively increasing order of difficulty through a sliding window approach. The approach achieves improvements on general instruction evaluation with GPT-2 (0.1B ) and on math/code tasks with Qwen2.5-1.5B.

**Strengths:**

The motivation and problem are practical. The plug-and-play module is very useful.

**Weaknesses:**

● Lack of originality: Using loss to measure instance difficulty and then applying curriculum learning is not a novel idea, as similar approaches have been widely explored in many curriculum learning-related papers (e.g., Self-paced Learning).
● Model and evaluation benchmarks: The model used (GPT-2) is relatively outdated, and for the general instruction-following evaluation, the paper does not adopt currently standard benchmarks such as IFEval, which are commonly used in the community.
● Writing: Some variables are unexplained, such as w_j in line 272. The variable definitions in the sliding window algorithm are unclear and can be confusing.

**Questions:**

The data introduced at each stage is predetermined based on the loss of the untrained student model. Is it reasonable to use the pre-defined loss to reflect the difficulty  for the current training model?

---

### Note · Authors · 2025-11-30

I have read and agree with the venue's withdrawal policy on behalf of myself and my co-authors.